# Role of healthcare cost accounting in pricing and reimbursement in low-income and middle-income countries: a scoping review

Lorna Guinness  ,[1] Srobana Ghosh,[1] Abha Mehndiratta  ,[1] Hiral A Shah  [1,2]

[1]Center for Global Development, London, UK
[2]Department of Infectious Disease Epidemiology, Imperial College London, London, UK

**Correspondence to**
Dr Lorna Guinness;
lornaguinness@gmail.com

## ABSTRACT

**Objectives** Progress towards universal health coverage (UHC) requires evidence-based policy including good quality cost data systems. Establishing these systems can be complex, resource-intensive and take time. This study synthesises evidence on the experiences of low-income and middle-income countries (LMICs) in the institutionalisation of cost data systems to derive lessons for the technical process of price-setting in the context of UHC.

**Design** A scoping review and narrative synthesis of publicly available information.

**Data sources** PubMed, MEDLINE, EconLit, the Web of Science and grey literature searched from January 2000 to April 2021.

**Eligibility criteria** English-language papers published since 2000 that identified and/or described development of and/or methods used to estimate or inform national tariffs for hospital reimbursement in LMICs. Papers were screened by two independent reviewers.

**Data extraction and synthesis** Extraction was performed by one reviewer and checked by the second reviewer on: the method and outputs of cost data collection; commentary on the use of cost data; description of the technical process of tariff setting; and strengths and challenges of the approach. Evidence was summarised using narrative review.

**Results** Thirty of 484 papers identified were eligible. Fourteen papers reported on primary cost data collection; 18 papers explained how cost evidence informs tariff-setting. Experience was focused in Asia (n=22) with countries at different stages of developing cost systems. Experiences on cost accounting tend to showcase country costing experiences, methods and implementation. There is little documentation how data have been incorporated into decision making and price setting. Where cost information or cost systems have been used, there is improved transparency in decision making alongside increased efficiency.

**Conclusions** There are widely used and accepted methods for generating cost information. Countries need to build sustainable cost systems appropriate to their settings and budgets and adopt transparent processes and methodologies for translating costs into prices.

## INTRODUCTION

Low-income and middle-income countries (LMICs) have been making significant progress towards universal health coverage (UHC) through innovative healthcare financing. One focus of healthcare financing reforms has been reimbursement schemes that target the explicit goals of efficiency and cost containment while improving quality and reaching the poor and vulnerable. Historically, block grants have been used to reimburse healthcare providers in publicly financed systems in LMICs. However, as national-level public purchasers have evolved and a broader range of healthcare providers (eg, private or faith-based healthcare providers) are accepted as part of the developing health system, newer prospective payment mechanisms and systems of provider reimbursement are being used by government purchasers of healthcare.[1]

Common prospective payment mechanisms such as case-based payments for the reimbursement of secondary and/or tertiary care and capitation payments for primary care providers are now being championed across developing regions and countries. Case-based payments are equivalent to a system where providers are reimbursed based on cases treated rather than per service or per bed days.[2] On the other hand, capitation-based

payments are equivalent to a payment system where lump sum payments are made to care providers based on the number of patients in a target population.[2 3]

### STRENGTHS AND LIMITATIONS OF THIS STUDY

⇒ The review relied on publicly available information and there may be further information that was not accessible.
⇒ The terminology on the role of cost evidence in price setting in the literature is poorly defined and inconsistent.
⇒ Our findings are limited by the inadequate documentation on how cost data and price evidence is used in the price-setting process in low-income and middle-income countries.
⇒ The review explored both costing methods used in cost systems and how cost information is used to inform price setting.

Setting reimbursement rates requires a reliable cost evidence base to enable price negotiations that are transparent, facilitate cost control and help drive providers to more efficient services. In principle, information is needed on the average cost per case across all admissions and/or visits (a base rate) and the relative value of different conditions as classified in the respective country (eg, diagnosis related groups (DRGs), specialty-based classification, intervention-specific health benefit package, etc).[4–6] In a case-based payment scheme, the service groups are often DRGs or a similar grouping system that provides a means of relating the type of patients a hospital treats to the costs incurred by the hospital. For capitation-based systems the grouping is related to the average expected cost of treating a patient under the care of the provider. In both types of systems, the technical process of price setting requires a robust cost system to be in place, using principles that can be guided or even mandated by a purchaser, in order to generate reliable health service cost estimates.

Raulinajtys-Grzybek[7] defines the cost system as *'a cost accounting system that ensure the cost homogeneity of individual groups* (of services)'.[7] There are however variations in costing systems across health systems as a result of choices about the process of collecting and verifying the data, the stage of development of the reimbursement system, the regulation around cost accounting and the costing methodology used.[7] For example, they can vary from one-off costing studies to regular national costing surveillance.[7 8] Some cost surveys involve all participating providers; for example, in the UK and Australia all providers are mandated to submit cost accounting information; in others, only a sample of representative providers is used, for example, France, Germany and Thailand.[1]

In terms of costing methodology, according to Gapenski and Reiter (2016) 'the holy grail of cost estimation is costing at the service or individual patient level'.[9 10] More advanced systems, for example, those in the UK and Australia use bottom-up style costing methods to derive patient-level DRG costs;[11] but there are simplified methods available that calculate the average cost of procedure through step-down allocation methods.[1] Whichever approach is taken, it is important that the costing is nationally acceptable and can capture structural differences in cost that might be present (types of provider, demography, geography, etc), as well as variability between the cost of the conditions treated. In addition, the national costing system should be standardised across providers, creating transparency and comparability.[8]

In LMICs, while the process of payment reform has been well documented, there is less information available about the role of cost information in the technical process of setting reimbursement rates. An increasing number of countries are moving towards case-based payment

schemes for secondary care within their UHC strategies. Documenting the cost systems used to generate evidence for rate setting can provide lessons for the further development of existing systems or the establishment of new ones. The aim of this paper is therefore to synthesise the evidence on the role of cost accounting in setting reimbursement rates for case-based payment schemes in LMICs. We performed a scoping review and narrative synthesis to document the current practice in LMICs based on publicly available information and recommend steps for the technical process of price setting in LMICS in the context of UHC goals.

## METHODS
### Search strategy and selection process
A scoping review approach was used to synthesise the evidence on cost accounting in LMICs. We aimed to map the body of literature, clarify key concepts and identify any gaps in the research.[12] We further refined our research question using a standard Problem, Intervention, Comparator, Outcome (PICO) framework:
► Problem: technical process for price setting for hospital case-based payments in LMICs
► Intervention: cost systems
► Comparator: non-cost-based methods
► Outcome: improved cost evidence base for price-setting

We used several approaches for identifying the literature. First, we conducted a search of the literature for peer-reviewed English-language publications indexed in PubMed, MEDLINE, EconLit and the Web of Science on the subject of national-level health system costing in LMICs and the associated design of their costing systems. Our search was conducted using the following terms: ('case*mix' or 'cost systems' or 'cost*accounting' or 'ref*-costs' or 'resource weights' or 'cost*weights' or 'national reimbursement' or 'DRG' or 'hospital payment systems' or 'fee*for*service') AND ('LMIC' or 'low resource settings' or 'developing countries'). We conducted a search that included the country name of all LMICs, as defined by the World Bank. To complement this, we consulted existing libraries of both grey and peer-reviewed literature held by the research team. We then conducted an analysis of text words contained in the title and abstract to help identify further keywords and index terms. A further search was then conducted using the identified keywords and index terms. Finally, the reference lists of all identified reports and articles were reviewed for any reports or papers that might have been missed. The search strategy is provided in the online supplemental table S1.

Papers in the English language were included. We searched for literature published between January 2000 and April 2021. We restricted the search to this time period as, in LMICs, case-based payments in national insurance programmes are a relatively new phenomenon and the quality and use of cost data were very limited prior to this.[13 14] Results were hand

**Identification**

Records identified from:
Econlit (n = 214)
Medline (n = 179)
Pubmed (n = 11)
Web of Science (n = 33)
Targeted review (n = 47)
**Total (n = 484)**

**Screening**

Records screened (title and abstract):
Econlit (n = 6)
Medline (n = 25)
Pubmed (n = 0)
Web of Science (n = 7)
Targeted review (n = 22)
**Total (n = 60)**

Records excluded:
Econlit (n = 208)
Medline (n = 154)
Pubmed (n = 11)
Web of Science (n = 26)
Targeted review (n = 25)
**Total (n = 424)**

**Included**

Studies included in review (after full text screening)
Econlit (n = 2)
Medline (n = 3)
Pubmed (n = 0)
Web of Science (n = 3)
Targeted review (n = 22)
**Total (n = 30)**

Reports excluded:
Econlit (n = 4)
Medline (n = 22)
Pubmed (n = 0)
Web of Science (n = 4)
Targeted review (n = 0)
**Total (n = 30)**

**Figure 1** Preferred Reporting Items for Systematic Reviews and Meta-Analyses (PRISMA) diagram.

screened to ensure that the topic was limited to the eligible countries (LMICs as defined by the World Bank) and that the study identified and/or described the development of the national tariffs for hospital reimbursements and/or the methods used to estimate or inform the tariffs for hospital services reimbursement. The titles and abstracts were screened independently by two reviewers (SG, LG) as per the inclusion and exclusion criteria defined by the study. The second screen involved reviewing full texts.

The papers were then classified according to whether they explained the technical process of price setting for reimbursements (ie, if and how cost data were used) and whether they reported on the process of primary cost data collection for price setting. For those papers or case studies reporting on the process

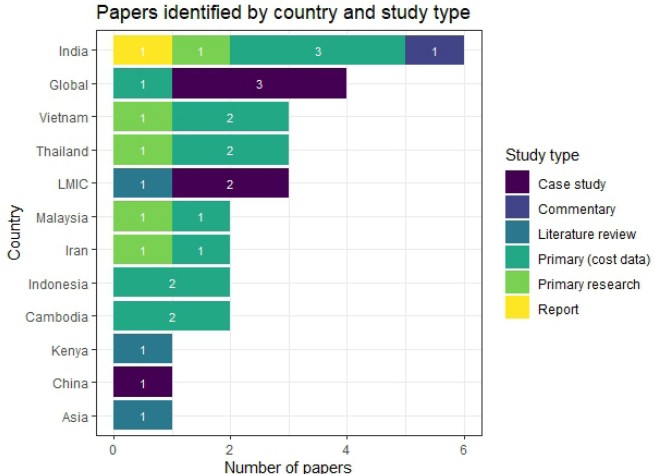

**Figure 2** Number of papers by country breakdown and types of study (some papers explore more than one country). LMICs, low-income and middle-income countries.

of primary cost data collection for price setting, we extracted information on the method of cost data collection, the output and any commentary on how the cost data were used for price setting for hospital case-based payments including identifying the commissioning agency. From the papers that described how cost data are used in price setting, we extracted information on any description of the technical aspects of the tariff setting system in place, at the time of the study, and the key strengths and challenges of the approach used. For those papers describing more than one country experience, only evidence on LMIC experiences was extracted. We use a narrative review approach to summarise the evidence by country. Data extraction was performed by one reviewer (SG) and then checked independently by another reviewer (LG).

### Patient and public involvement

Neither patients nor the public were involved in the design, conduct, reporting or dissemination plans of our research.

### RESULTS
### Overview of the literature

A total of 484 papers was initially identified of which 424 papers were excluded in the initial screening. The second screen involved reviewing full texts, leading to the inclusion of 30 papers in the review as described in the Preferred Reporting Items for Systematic Reviews and Meta-Analyses (PRISMA) diagram in figure 1.

Of the 30 papers extracted (see online supplemental table S2), 7 papers stated a global focus (including LMICs)[1 4 15–19] and one paper reported to be focused on the Asia region[20] (see figure 2). Of the single country focused papers, six related to India.[21–26] We found three studies each related to Thailand[6 27 28] and Vietnam.[29–31] There were two studies focused on each of Indonesia,[32 33] Iran,[34 35] Malaysia[36 37] and Cambodia[38 39] and one study each for Kenya[40] and China.[41] Further, within the global papers, we identified case studies on: Kazakhstan, Kyrgyzstan, India, Malaysia, Thailand and China.[1 4 15–19]

### Papers reporting on primary collection of cost data to inform tariff setting

Twenty-three case studies from 14 studies reported on primary cost data collection for price setting purposes in a single country setting, either describing methods or both methods and results (see online supplemental table S3). Twelve case studies also had the explicit aim of generating cost information for broader policy processes. In terms of pricing, two case studies reported on a costing exercise that was designed to inform capitation payment rates,[17 32] six studies aimed at generating cost weights for DRGs[4 17 37] (Joint Learning Network case studies: Central Asian Republics) or unspecified case groups[17 28 30] (Joint Learning Network case studies: Indonesia Ministry of

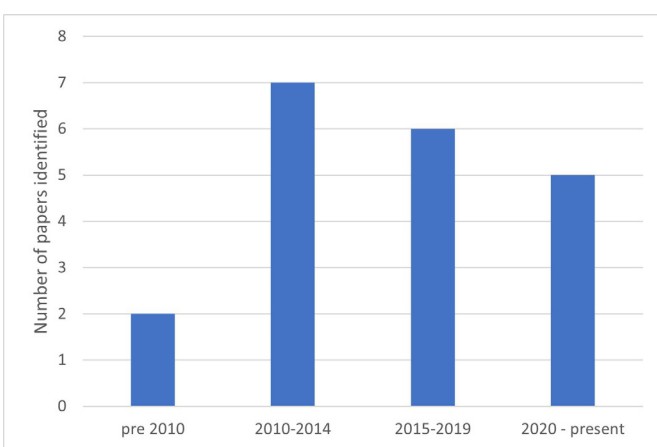

**Figure 4** Number of papers explaining the tariff-setting scheme by year of publication.

Health) and three studies reported on the estimation of the costs of health benefit packages (Joint Learning Network case studies: PhilHealth and India Aarogyasri).[17 26] A final case study reviewed the available cost evidence for informing price setting in the National Health Insurance Fund, Kenya.[40]

For the studies reporting costs, cost per service unit at the hospital level was the most frequently sited output, for example, cost per bed day, cost per admission and cost per outpatient visit. Three studies generated unit costs for specific services: cost per adverse event,[27] laboratory services[33] and pharmacy services.[37] A further three case studies generated costs of health benefit packages (Joint Learning Network case studies: PhilHealth and India Aarogyasri).[17 26] Relative value units (RVU's) were the primary output of seven studies,[17 25 28 30 31 35 37] one of which also explicitly estimated an inpatient base rate.[28]

Fourteen of the case studies were commissioned by the local ministry of health or agency acting on their

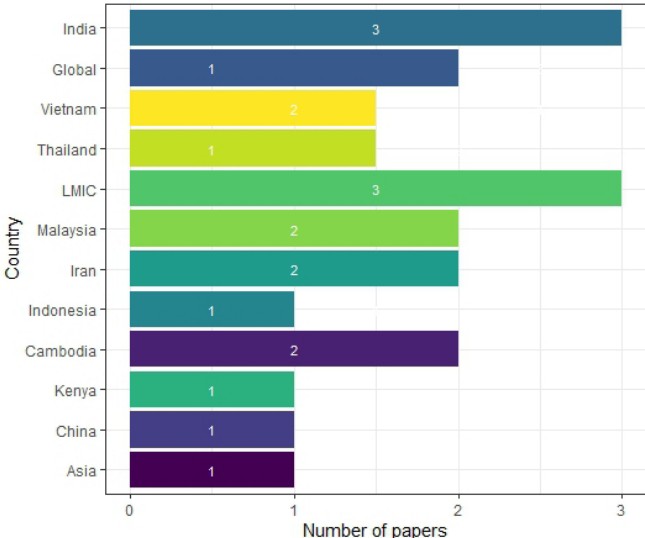

**Figure 3** Number of papers explaining the tariff-setting scheme by country. LMICs, low-income and middle-income countries.

behalf. However, in many cases, it was not clear who had commissioned the costing or if the study was linked to the national policy process[15 17 24 25 27 30 31 37] (Joint Learning Network case studies: India – Public Health Foundation of India). Two studies evaluated different methods for generating robust RVUs.[31 35]

### Papers reporting on how cost data informs the tariff-setting process

We identified 18 papers that provided explanation of the technical process of tariff setting, documenting experiences in 10 different countries (see figure 3). Five papers were published since the beginning of 2020, six papers in the period 2015–2019, 7 papers in the period 2010–2014 and two papers were published before 2010 (see figure 4). The papers provided mixed levels of detail on the technical processes of price setting and the strengths and weaknesses in each locality.

The current tariff system, presence of an explanation of the price-setting process, the data used in price setting, and resulting policy levers and implications are summarised by country in figure 5. Only one study described tariff setting in Africa.[40] The paper reviewed the available evidence on costs for informing Kenya's National Health Insurance Fund prices and was published in 2011. Although the cost information were considered reliable by all stakeholders, in part due to their involvement in the costing exercises, the costs had not been used for setting prices at the time of the study.

The other countries covered were all in Asia. In the Central Asia region, three papers focus on the reform of the tariff setting system in Kyrgyzstan.[4 16 19] The technical process of price setting is clearly documented. This process includes cost control measures derived from linking the reimbursement rates to the ministry of health budget. The Thai Universal Coverage Scheme (UCS) also provides an example in which cost control is built into the base rate through linkages to the budget. However, as figure 5 states, in Thailand, there are three government funded and implemented schemes. Although all Thai schemes use the same Thai-DRG grouper, the Civil Service Medical Benefit Scheme and Social Health Insurance schemes do not use cost control mechanisms, as the rates are not linked to an overall budget and are different rates for different hospitals.[1 6 19 28]

The Kyrgyz reforms were found to be vulnerable to gaming as the system does not make full use of the data available potentially leading to misclassification of diagnoses. This potential for gaming is also highlighted in Iran. In contrast to the systematic introduction and use of cost accounting in Kyrgyzstan, Iran's price-setting process involves a technical assessment by an independent body but with limited transparency (see figure 5).[34 35] Doshmangir *et al* note that without an objective and explicit mechanism in the updating of medical tariff and no structure to effectively manage conflicts of interest, the pricing system has in effect become 'a tool for revenue manipulation'.[34]

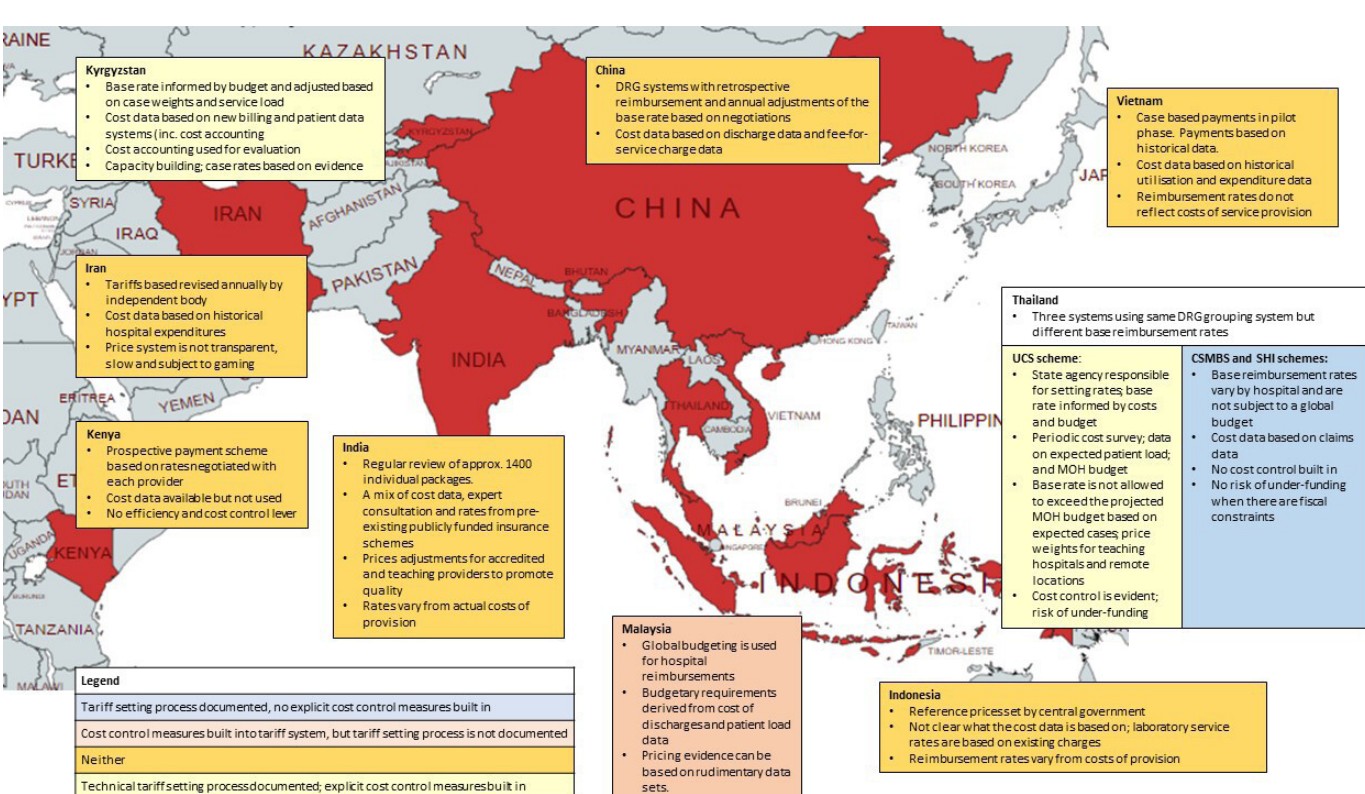

**Figure 5** Summary of evidence on the tariff setting process for case-based hospital payment in national health insurance schemes. CSMBS, Civil Service Medical Benefit Scheme; DRG, diagnosis related groups; MOH, Ministry of Health; SHI, Social Health Insurance; UCS, Universal Coverage Scheme.

Challenges also arise when reimbursement rates are not based on cost evidence. In India's national insurance programme for the poor and vulnerable, the government used existing information to set reimbursement rates while establishing a review system to allow for modification and improvement over time (see figure 5).[16] However, the method in which information from costing studies, experts and rates under previous schemes is compiled is not transparently reported on. Studies show that the rates vary considerably from actual costs (42% of (Health Benefit Packages (HBPs) had a price less than 50% of the true cost in 2018).[26] This could affect the recruitment of providers, the coverage and quality of care, and bring the rates themselves into doubt (Prinja S *et al* 'Determining Price Weights for Differential Case-Based Payments under India's National Publicly Financed Health Insurance Programme', Unpublished, 2022). Indonesia faces a similar problem in respect of laboratory services. Dianingati *et al* report on a lack of transparency in the development of the reference prices set by the government and that the true costs of service delivery are 40%–53% higher than the reference price.[33] However, validation of the rates is difficult as data on the cost of healthcare services is still limited to a few services in focal geographical areas, restricted to the public sector with few published and readily accessible cost data analyses/data sets.[21 26]

Both the Thai UCS and Kyrgyz price-setting systems use cost accounting to inform their base rates and case weights. Langebrunner *et al* note how cost accounting has been used as an evaluation tool and allowed for tariff adjustments based on evidence so that payments match services in Kyrgyzstan.[4] Similarly, in the Thai UCS scheme, a key feature of the tariff setting is the cost information on which pricing for UCS is based. This is collected on a periodic basis in a cost survey and has evolved from initial work using an RVU method and 'top level' hospital cost data[28] to a 900 hospital survey. While no study described how the system has reformed, the papers note that the gradual, stepwise implementation allowed for institutional and technical capacity building.

There was less detail reported on the tariff-setting process in China, Indonesia, Malaysia and Vietnam. One study documenting tariff-setting processes in China raises concerns that the schemes used a retrospective payment system and failed to build in efficiency and cost control.[19] In Vietnam there was also concern that the fees and the payment schemes bore little relationship to the costs of delivering services, although the RVU method used to calculate rates for the capitation scheme was relatively simple. Similarly, while the Malaysian system was designed for global budgeting, it also demonstrates that pricing evidence can be based on skeletal data sets such as those that focus on large expenditure items and patient data that are feasible to collect.[1]

## DISCUSSION
### Key findings
Our scoping review has explored the literature on using the cost evidence base for setting prices in national health insurance schemes in LMICs. It has identified a significant gap in the literature in this area. However, despite this, we found consistent themes around the need to use cost information using a systematic methodology, reporting this transparently and working with providers to develop the system. Our review confirms that cost evidence can increase efficiency of service provision by increasing the policy evidence base. To generate this evidence, countries need to build cost systems appropriate to the setting and data availability but allowing for and investing in increasing complexity as data systems improve. While national costing surveillance should be an aspiration, prices may be set using cost evidence from one-off costing studies, or even hospital charges. The method in which these data are then used to set base rates and price weights, especially in the absence of national surveillance, should be part of a transparent process that involves relevant stakeholders and takes account of heterogeneity in costs driven by demand-side (eg, condition or patient-specific) and supply-side (eg, hospital location) factors.

### Strengths and weaknesses of the scoping review
The scoping review found a limited level of evidence and many of the studies are old and may be outdated. In addition, early reforms were reported for some countries, and it was not possible to determine how the tariff-setting processes have evolved. For example, Vietnam's pilot study was published in 2014 but there were no corresponding papers documenting the next steps; nor did we identify more recent reports on Kenya and Cambodia where costing evidence from large multisite studies were identified.

A second limitation was the focus of the review on a single component of the tariff-setting process which may have limited the evidence generated. During the screening process many articles were identified on the process of developing DRG-type reforms, but few focused on the price-setting process and how cost evidence is used in the price-setting process. This was compounded by the terminology related to the role of cost evidence in price setting in the literature which is poorly defined. The terms cost and price are used in many different ways to mean different things which may have led to some omissions. We addressed these issues by extending the search and performed additional searches using the keywords identified in the initial papers found, that met the inclusion criteria. Finally, our review of the grey literature was limited to a Google search and snowballing from references that were identified in the initial search. It is likely that evidence in this area lies in government and donor reports that we missed and restricting reports to the English language may have compounded this.

These challenges serve to highlight the lack of attention on this aspect of tariff setting in the literature and the need for further research.

### Implications of the findings for researchers and policy makers keen to establish cost systems
The key message from the review is that cost systems help create a transparent evidence-based process for price setting. A centralised cost accounting system, such as that developed in Kyrgyzstan, was considered a major strength of the broader health system reforms—allowing for policy reform to anticipate expenditure needs and enabling the government to effect change more effectively. Leaving base rates open to negotiation at the individual provider level, with minimal evidence on costs and efficiency of service provision, leaves the system vulnerable to gaming. Studies from Iran and Thailand emphasise how important the cost system is in the setting of health benefit package/DRG prices, to minimise gaming and prevent cost escalation.[6 34]

In addition, creating a tariff setting system that does not use costs based on empirical evidence can embed inefficiencies and possibly make it more difficult to implement costing in the future,[34] further underlining the need for cost systems to generate good quality data, based on accepted methodologies. As well as cost data collection, a systematic method for translating costs into prices or reimbursement rates helps avoid skewed incentives within the prices, evident in the unexplained differences between costs and reimbursement rates found in India and Indonesia.[26 33] Langenbrunner's reporting of the Kyrgyzstan case study provides the most comprehensive description for the calculation of the base rate and case-based weightings and how to use these to set reimbursement rates.[4] Patcharanarumol et al also describe the principles applied for estimating the base rates and weights in the Thai UCS scheme.[6] For both settings, explicitly accounting for the budget in the price estimation using an 'economic adjustment' is a key mechanism of cost control. This level of transparency is not apparent elsewhere in the literature identified. For example, while the different strands of information used for price setting are documented in the reports on India, the method for combining this information is not available.[26 42] In Iran, the lack of such a methodology was reported as a significant problem for the DRG system as a whole leading to price manipulation by different stakeholders.[34]

Methods for cost data collection also need to be appropriate for the setting. Studies from Thailand and Vietnam compare different approaches to obtaining the base rate and cost weights for health technology assessment and pricing. They compare microcosting with RVU approaches and find both to be feasible with microcosting being highly resource intensive. The costing methods tend to follow the same principles using top-down allocation methods supplemented with bottom-up costing if resources allow, for some specific inputs. In Malaysia, one study demonstrated the feasibility of using the electronic

prescribing system to generate DRG weights, although it was recognised that these were not available in most facilities.

It is also important to be aware of the trade-off in accuracy and resources needed to generate the required cost information. Where resources are highly constrained, any data can be better than no data, particularly if the data are reported transparently and how the data inform decisions is clearly communicated and accounted for. If cost accounting is not the norm and the budget is limited, costing for price setting may need to start with simpler methods, using, for example, expenditure data, RVUs and smaller samples of facilities. Alternative approaches, for example, in India, Cambodia and Kenya, have started with the implementation of baseline multisite costing studies. Although these are one-off exercises, they provide an evidence base and good practice on which to build. The costing itself can also be a way to bring stakeholders into the price-setting process and build capacity for future costings. The example of Kyrgyzstan shows how implementing a cost system is a slow, gradual and complex process. The established costing systems identified in the literature illustrate how a cost system has evolved from one-off exercises and developed into a complex system with increasing numbers of participating providers (Thailand, Kyrgyzstan, China).

### Future research

Our review is the first study in this area for LMIC settings, providing a foundation on which further evidence in this area can be developed. More work is required to document better the practice of cost data collection, the costing methods used for informing national tariffs and how cost information is integrated into the tariff-setting process to guide future reforms in health system financing within LMICs.

### CONCLUSION

LMICs are increasingly turning to insurance-based models of healthcare and to private sector providers to increase coverage of the poor and vulnerable. To help achieve value for money within these UHC goals, publicly financed insurance schemes need to account for budget constraints, encourage efficient health service delivery and use good quality evidence transparently in setting reimbursement rates. Documentation of the good practice and the challenges of generating cost evidence and creating costing systems for informing reimbursement decisions in resource-poor settings is lacking. While there are accepted and widely used methods for generating cost information, countries need to build more sustainable cost systems and adopt more transparent systems and methodologies for translating costs into prices.

**Acknowledgements** The authors thank Y-Ling Chi and Lydia Regan at the Center for Global Development for valuable comments and advice on the search strategy.

**Contributors** LG, HAS, SG, AM collaborated on the concept and design. Data extraction, analysis and interpretation were carried out by LG and SG. The manuscript was drafted by LG, SG and HAS. Critical revisions of the paper for important intellectual content were made by LG, SG, HAS and AM. Funding was obtained by LG and AM and supervision was carried out by AM. LG is guarantor.

**Funding** This work was supported by the Bill & Melinda Gates Foundation Grant no: INV-003239.

**Map disclaimer** The inclusion of any map (including the depiction of any boundaries therein), or of any geographical or locational reference, does not imply the expression of any opinion whatsoever on the part of *BMJ* concerning the legal status of any country, territory, jurisdiction or area or of its authorities. Any such expression remains solely that of the relevant source and is not endorsed by *BMJ*. Maps are provided without any warranty of any kind, either express or implied.

**Competing interests** HS contributed to this study while being employed for the Center for Global Development. HS is now an employee for GSK and holds shares in the GSK group of companies. LG declares fees for postgraduate teaching at the London School of Hygiene and Tropical Medicine. AM and SG declare no competing interests.

**Patient and public involvement** Patients and/or the public were not involved in the design, or conduct, or reporting, or dissemination plans of this research.

**Patient consent for publication** Not applicable.

**Ethics approval** Not applicable.

**Provenance and peer review** Not commissioned; externally peer reviewed.

**Data availability statement** Data sharing is not applicable as no data sets were generated and/or analysed for this study.

**ORCID iDs**
Lorna Guinness http://orcid.org/0000-0002-1013-4200
Abha Mehndiratta http://orcid.org/0000-0003-3045-1649
Hiral A Shah http://orcid.org/0000-0003-0204-451X

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
