## [Reviewer comments · BMJ Open]

ARTICLE DETAILS

TITLE (PROVISIONAL)	The role of healthcare cost accounting in pricing and reimbursement in low- and middle-income countries: a scoping review
AUTHORS	GUINNESS, LORNA; Ghosh, Srobana; Mehndiratta, Abha; Shah, Hiral A.

VERSION 1 – REVIEW

REVIEWER	Obinna Ikechukwu Ekwunife Nnamdi Azikiwe University, Clinical Pharmacy and Pharmacy Management
REVIEW RETURNED	22-Jun-2022

GENERAL COMMENTS	This is an important scoping review that summarizes the role of cost accounting in setting reimbursement rates for case-based payment schemes in LMICs. This is an important study and will benefit countries that want to set healthcare cost accounting systems. I have some minor comments for the authors: Was there any protocol for the scoping review and was it published anywhere? Page 3 of 35 – Line 12, add “to” in “...countries (LMICs) in trying institutionalize...” Page 5 of 35 – Line 49, specify the meaning of the acronym DRG in the first instance Page 6 of 35 – Line 57. Was there any rational form of searching for only studies from 2000? If so, it should be included. Page 7 or 35 – it was not clear how many authors reviewed the full texts. It may be a good practice to state the initials of the authors that did the screening of title & abstracts and review of full manuscript. Page 8 of 35 – Line 38. The sentence should read “Data extraction was performed by one reviewer and then checked independently by another reviewer”. As stated earlier, it will be good to specify the author that did the task using their initials. Page 9 of 35 – Line 8. Correct focussed to “focused” Page 12 of 35 – First paragraph in the discussion section should be dedicated to summarizing the major finding of the study Figure 5 – On Thailand – Correct the word “grouper”
---

REVIEWER	Anelisa Jaca South African Medical Research Council
REVIEW RETURNED	16-Aug-2022

GENERAL COMMENTS	Thank you so much for a well written and informative paper. I just have minor comments. Methodology
--

	What is the rationale for only including papers published since 2000? Could you elaborate on "improved evidence base for decision-making" in the methods section?
--	---

VERSION 1 – AUTHOR RESPONSE

Reviewer: 1	
This is an important scoping review that summarizes the role of cost accounting in setting reimbursement rates for case-based payment schemes in LMICs. This is an important study and will benefit countries that want to set healthcare cost accounting systems. I have some minor comments for the authors:	Thank you.
Was there any protocol for the scoping review and was it published anywhere?	Thanks for this question. No protocol was published.
Page 3 of 35 – Line 12, add “to” in “...countries (LMICs) in trying institutionalize...”	Thank you – this has been corrected
Page 5 of 35 – Line 49, specify the meaning of the acronym DRG in the first instance	Thank you for spotting this; we have now specified the meaning 2 lines above when “Diagnosis Related Groups” is first introduced. Pg 4, 2 nd para, line 5
Page 6 of 35 – Line 57. Was there any rational form of searching for only studies from 2000? If so, it should be included.	We have now added the rationale for this: “We searched literature published between January 2000 and April 2021. We restricted the search to this time period as in LMICs case based payments in national insurance programmes are a relatively new phenomenon and the quality and use of cost data was very limited prior to this [14,15].” Pg 6-7
Page 7 or 35 – it was not clear how many authors reviewed the full texts. It may be a good practice to state the initials of the authors that did the screening of title & abstracts and review of full manuscript.	This is a good point – thank you. We have now added the initials of the reviewers where relevant. Pg 7
Page 8 of 35 – Line 38. The sentence should read “Data extraction was performed by one reviewer and then checked independently by another reviewer”. As stated earlier, it will be good to specify the author that did the task using their initials.	Thank you for this correction; we have edited this sentence as suggested. Pg 7
Page 9 of 35 – Line 8. Correct focussed to “focused”	Thank you for spotting this – we have now edited this throughout the document.
Page 12 of 35 – First paragraph in the discussion section should be dedicated to summarizing the major finding of the study	Thank you for this suggestion we have now made this edit to the discussion. Pg 11.
Figure 5 – On Thailand – Correct the word “grouper”	Thank you for this – we have changed the word “grouper” to “DRG grouping system” in figure 5.
Reviewer: 2	
What is the rationale for only including papers published since 2000?	We have now added the rationale for this: “We searched literature published between 2000 and April 2021. We restricted the search to this time period as in LMICs case based payments in national insurance

	programmes are a relatively new phenomenon and the quality and use of cost data was very limited prior to this [14,15].” Pg 6-7
Could you elaborate on "improved evidence base for decision-making" in the methods section?	Thank you for this comment. We agree that this could be specified more clearly and have corrected it as follows: “improved cost evidence base for price-setting” Pg 6, last bullet of PICO framework

VERSION 2 – REVIEW

REVIEWER	Obinna Ikechukwu Ekwunife Nnamdi Azikiwe University, Clinical Pharmacy and Pharmacy Management
REVIEW RETURNED	13-Sep-2022
GENERAL COMMENTS	The reviewer completed the checklist but made no further comments.